# Gut microbiota composition of the isopod *Ligia* in South Korea exposed to expanded polystyrene pollution

Young-Mi Lee[1], Kwang-Min Choi[2], Seong Hee Mun[2], Je-Won Yoo[1], Jee-Hyun Jung[2,3]*

**1** Department of Biotechnology, College of Convergence Engineering, Sangmyung University, Seoul, Republic of Korea, **2** Risk Assessment Research Center, Korea Institute of Ocean Science and Technology, Geoje, Republic of Korea, **3** Department of Marine Environmental Science, Korea University of Science and Technology, Daejeon, Republic of Korea

* jungjh@kiost.ac.kr

**Data Availability Statement:** All data are in the manuscript and/or Supporting information files.

**Funding:** This work was supported by the grants "Risk assessment to prepare standards for protecting marine ecosystem (KIMST-20230383)",

## Abstract

Plastics pose a considerable challenge to aquatic ecosystems because of their increasing global usage and non-biodegradable properties. Coastal plastic debris can persist in ecosystems; however, its effects on resident organisms remain unclear. A metagenomic analysis of the isopoda *Ligia*, collected from clean (Nae-do, ND) and plastic-contaminated sites (Maemul-do, MD) in South Korea, was conducted to clarify the effects of microplastic contamination on the gut microbiota. *Ligia* gut microbiota's total operational taxonomic units were higher in ND than in MD. Alpha diversity did not differ significantly between the two *Ligia* gut microbial communities collected from ND and MD, although richness (Observed species) was lower in MD than in ND. Proteobacteria (67.47%, ND; 57.30%, MD) and Bacteroidetes (13.63%, ND; 20.76%, MD) were the most abundant phyla found at both sites. Significant different genera in *Ligia* from EPS-polluted sites were observed. Functional gene analysis revealed that 19 plastic degradation-related genes, including those encoding hydrogenase, esterase, and carboxylesterase, were present in the gut microbes of *Ligia* from MD, indicating the potential role of the *Ligia* gut microbiota in plastic degradation. This study provides the first comparative field evidence of the gut microbiota dynamics of plastic detritus consumers in marine ecosystems.

## Introduction

Plastics are a great concern in aquatic ecosystems because of their increasing global usage and worldwide distribution by non-biodegradable properties [1]. Plastic debris is decomposed into smaller particles, namely, microplastic (MP, < 5 mm) and nanoplastic (< 1 μm) via physical (such as UV radiation and weathering) and biological reactions in natural environments [2]. MPs can accumulate in the gut of aquatic organisms and biomagnify to higher tropical levels, leading to adverse effects in humans [3, 4]. Several studies have suggested that MPs cause oxidative stress and immune toxicity that exert deleterious effects on the growth, development, and reproduction of aquatic invertebrates [5–8].

and "Land and sea based input and fate of microplastics in the marine environment (KIMST-20230383) of the Korea Institute of Marine Science & Technology Promotion (KIMST) funded by the Ministry of Oceans and Fisheries", and the National Research Foundation of Korea (NRF) grant funded by the Korea government (MSIT) (No. 2023R1A2C1005630) The funders had no role in study design, data collection and analysis, decision to publish, or preparation of the manuscript.

**Competing interests:** The authors declare that they have no competing financial interests or personal relationships that may have influenced the work reported in this study.

MPs can be ingested by invertebrates, such as polychaetes, bivalves, crustaceans, and fish [9–11]. Avio et al. [12] reported that Mediterranean mussels (*Mytilus galloprovincialis*) exposed to polyethylene (PE) and polystyrene (PS) MPs contained plastic aggregates in their intestinal lumen, epithelium, and tubules. Ingested plastic particles can cause direct damage, resulting in intestinal dysbiosis [13]. Several studies have demonstrated that exposure to MPs can alter gut bacterial diversity and exert negative health effects, such as inflammation [14–16]. More recently, Tamargo et al. [17] reported that polyethylene terephthalate (PET) MPs changed the intestinal microbiota communities in humans during the fermentation of ingested MPs. These studies supported that MP exposure may change the composition and function of the intestinal microbial communities. 16S rRNA gene amplicon sequencing has been widely used to identify microbial communities [18]. More recently, owing to advancements in high-throughput sequencing, metagenomic analyses have provided better insights regarding the dynamics, function, and interaction of microbial communities with environmental pollution in ecotoxicological studies [19, 20].

The digestive tracts of arthropods, such as insects, perform specific functions in the host via interactions between microbiome constituents [21]. In particular, collaboration between insect gut microorganisms plays a key role in the degradation of polypropylene (PP), PE, and PS [22]. A recent study has shown that the plastic-specific gut microbiota increased in the isopod *Porcellio scaber* exposed to PS and PET, indicating the role of the isopod gut microbiome in plastic digestion [23]. These studies suggested that the digestive tracts of isopods, as well as insects, may be important sources of microbiomes with novel and effective plastic degradation enzymes. However, little information on gut microbiome in the isopods is available, in particular, along their habitat.

*Ligia* (Crustacea: Isopoda: Ligiidae) is a large gray-brown isopod, that inhabits the intertidal regions of coastal areas. They play an ecologically important role as scavengers, feeding mostly on detritus and plant debris (including plastic, plants, and chemical pollutants) in coastal areas during nutrient cycling in marine ecosystems [24–26], rendering them suitable species for studying the correlation between ingested MPs and the gut microbiota of resident organisms in coastal environments. In particular, as *Ligia* can accumulate organotins (such as tributyltin) [25], polycyclic aromatic hydrocarbons [27], heavy metals [28], MPs, and plastic additives [26, 29], they are considered excellent pollution biomonitoring organisms in coastal areas [30].

In the present study, to investigate the microbiome composition and identify functional genes, 16S rRNA amplicon sequencing, and metagenomics shotgun sequencing were performed in *Ligia* collected from the reference site and plastic-contaminated sites in South Korea, respectively. Our results are expected to improve our understanding regarding the impacts of expanded polystyrene (EPS) ingestion on the microbiome of isopods and suggest a potential role of *Ligia* in plastic degradation.

## Materials and methods

### Experimental species

The sampling locations of *Ligia* are shown in S1A Fig. *Ligia* spp. (body length: 23.02 ± 1.17 mm; body weight: 0.70 ± 0.12 g) were collected at Nae-do (ND; 34˚38'47.30"N 128˚34'39.00"E; Clean site, CS) and Maemul-do (MD) (34˚47'21.50"N 128˚42'58.10"E; Polluted site, PS). The location was classified according to the degree of plastic pollution based on Choi et al. [26] and this area was an area where there was no permitting procedure required for marine *Ligia* spp. collection; thus, it was legal to freely sample. The collected *Ligia spp.* were dissected and sampled for sequencing of each hepatopancreas, following Cho et al. (2019) [31] and Choi et al. (2023) [26]. The hepatopancreas was digested using 10% potassium hydroxide solution, sieved

using a 20 μm sieve, and transferred to a lithium meta-tungstate solution (LMT, 1.6 g/cm$^3$) in a funnel.

## Identification of the collected *Ligia* spp.

For phylogenetic analysis of *Ligia* spp. collected from ND and MD, the mitochondrial 16S rRNA region (approximately 490 bp) was amplified using a published primer set (16Sar: 5′-CGCCTGTTTATCAAAAACAT-3′ and 16Sbr: 5′-CCGGTCTGAACTCAGATCACGT-3′) [32, 33]. The polymerase chain reaction (PCR) mixtures consisted of 10 ng of genomic DNA, 1 × PCR buffer, 0.2 mM dNTP, 0.5 mM of each primer, 1 × GC enhancer, and 0.05 unit/μL of Axen™ *Taq* DNA polymerase (Macrogen; Seoul, Korea). PCR was performed in a T100™ thermal cycler (Bio-Rad Inc., Hercules, CA, USA) under the following conditions: 94 °C for 4 min, followed by 35 cycles at 94 °C for 40 s, 45 °C for 40 s, and 72 °C for 40 s, and a final elongation at 72 °C for 4 min. The PCR products were sequenced and used for phylogenetic analysis. The 16S rDNA sequences of 25 *Ligia* spp. (*L. exotica*, *L. cinerascens*, *L. oceanica*, *Ligia* spp. eastern groups, and *Ligia* spp. western groups) were retrieved from GenBank and aligned with our sequences using Geneious Prime (version 2022.2.2). After alignment, the ambiguously aligned regions were eliminated using GBLOCKS v.0.91b with the default parameters [34]. Subsequently, a phylogenetic tree of the aligned sequences was constructed using MEGA (version 11.0.13) and the neighbor-joining method with the Kimura 2-parameter model.

## DNA extraction and sequencing

DNA was extracted from the *Ligia* hepatopancreas using a QIAamp PowerFecal Pro DNA kit (Qiagen, Hilden, Germany). All the extracted DNA samples were analyzed using a NanoDrop spectrophotometer. Only samples that met the qualitative criteria ($A_{260}/A_{280}$ ratio = 1.8–2.0) were included in the analysis. High-throughput sequencing was performed by HME Healthcare Co. Ltd. (South Korea). Libraries for the 16S rRNA gene (V3-V4 region) and metagenome shotgun sequencing were constructed using Illumina Novaseq 6000. 341F (CCTAYGGGRBGCASCAG) and 806R (GGACTACNNGGGTATCTAAT) primers were used for V3-V4 sequencing. Metagenome shotgun sequencing libraries were constructed with insert sizes of 400 bp using an Illumina TruSeq Nano DNA LT library preparation kit. Each library was sequenced using the Illumina NovaSeq platform (Illumina, USA), with at least 10 G data per sample.

## Bioinformatics analysis

**16S rRNA sequencing data.** The raw 16S rRNA V3-V4 sequencing data were analyzed using QIIME2, with a sequence denoising step of DADA2. Operational taxonomic units (OTUs) of 97% reads were selected *de novo* and the Silva database (Silva release 138) was used to annotate the classification of OTUs. A phylogenetic tree was created by aligning the representative sequences using MAFFT, masking them using an alignment mask, and building trees using FastTree in QIIME2. Bacterial alpha diversity was evaluated based on the observed species, Shannon and Simpson 1-D diversity indices, and the Chao-1 richness index using the estimated richness function of the R package, phyloseq. Statistical differences in the alpha diversity indices were tested using the Wilcoxon rank-sum test. The microbial community structure was analyzed using Bray–Curtis distances based on taxon relative abundance data. Principal coordinate analysis (PCoA) was performed as a multivariate unsupervised data exploration using vegan and visualized using ggplot2. Linear discriminant analysis (LDA) effect size (LEfSe) (LDA value > 3) was used to determine between-group differences, combined with the Kruskal–Wallis test ($p < 0.05$).

**Metagenome sequencing data.** The qualified reads were trimmed, filtered using Trimmomatic, and subjected to MetaSPAdes prior to assembly. Reads matching small rRNA genes, including 16S and 18S rRNA genes, in each sample were identified using PhyloFlash and re-assembled using Spades. All small rRNA genes were classified using the SILVA database (release 138) with the last-common-ancestor algorithm and aligned using MAFFT. The alignments were subjected to IQ-TREE for phylogenetic analysis. MetaPhlAn was used to quantify the relative abundance of taxonomic groups in each sample. MetaWRAP was used to bin genomes from assemblies with different modules, including metabat2, maxbin2, and Concoct. The Bin_refinement module (-c 50 - × 10 options) of MetaWRAP was used to consolidate all bins from different modules into a final set of bins. The quality of the genome bins was evaluated using the CheckM software. To classify all genome bins, GTDB-Tk v1.5.0 was used with R95, the most updated database. All qualified reads were mapped to de-replicated bins, and coverage was evaluated to calculate the relative abundance of each genome bin using coverM (https://github.com/wwood/CoverM). Genes with lengths greater than 500 bp were called from all contigs of each sample using Prodigal (-p meta). The reads were aligned using BWA-MEM for all predicted genes. Pileup.sh from the BBmap suite v37.25 was used to calculate the reads per kilobase per million mapped reads and number of reads mapped to each gene (BBMap short read aligner, http://sourceforge.net/projects/bbmap). Database alignment and functional annotation were conducted using the Kyoto Encyclopedia of Genes and Genomes (KEGG) and Plastic Biodegradation database (PlasticDB, https://plasticdb.org). Differentially abundant genes between the two groups were identified using the R package DESeq2 v 1.34. Bray–Curtis dissimilarities among samples were visualized using non-metric multidimensional scaling (vegan R package).

# Results and discussion

## Characterization of the collection sites, ND and MD

Studies on the distribution of plastic in the 20 coastal seashores of South Korea revealed that fragment type MP, with size ranging from 100–150 μm, was abundant, and that EPS occupied 95% of large-sized MPs (1–5 mm) [35]. MD and ND are situated off the southern coast of Korea and have similar hydrodynamic conditions (e.g. active water circulation and semi-diurnal tidal cycle), topographies, marine biotic communities (particularly in the upper and lower intertidal zones), limited anthropogenic pressures (such as no industrial or aquaculture activities), and limited populations (MD: 0.086 capita/km$^2$; ND: 0.046 capita/km$^2$), but considerably different marine debris pollution levels [29].

In our previous study, a significantly higher amount of marine plastic debris (78 ± 39 items/m$^2$ and 1730 ±1250 g/m$^2$) was detected in the upper intertidal zone of the MD, while its level was statistically lower at the ND (0.5 ± 0.3 items/m$^2$ and 2.6 ± 2.2 g/m$^2$). EPS was the major constituent among various plastic debris items found in the MD (79% by number, 32% by weight) [29]. Consistently, the body abundance rate of MPs (> 20 μm) was higher in *L. exotica* from MD (50.56 particles/individual) than in those from ND (1.00 particles/individual) [26] (S1B and S1C Fig). EPS (93.3%) was the most dominant polymer type in the *Ligia* samples from MD [26]. The MP concentration in *L. exotica* from MD was higher than those reported in other marine organisms, including the Manila clam (mean values of 0.43 ± 0.32 n/g and 2.19 ± 1.20 n/individual) and oysters in the Korean coastal area (1.21 ± 0.68 n/individual) [36]. The high body burden of MPs indicated that the coastal environment of the MD, including the sediment, water, and biota, was considerably more polluted by plastic litter than that of the ND [26]. These findings demonstrate that *Ligia* from MD is highly contaminated with environmental EPS via ingestion.

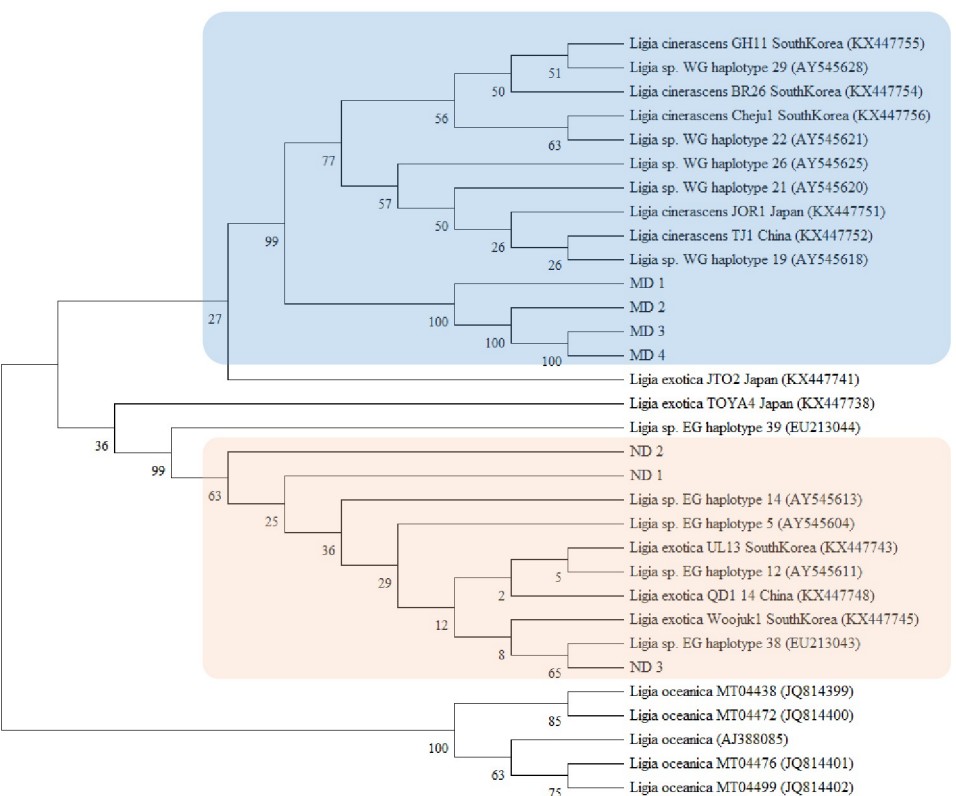

**Fig 1. Phylogenetic relationship of *Ligia* spp. from Nae-do (ND) and Maemul-do (MD) using 16S rRNA sequences.** Sequences of other *Ligia* spp. were retrieved from GenBank. *L. oceanica* was used as the out-group. The phylogenetic tree of the aligned sequences was constructed using the neighbor-joining method with the Kimura 2-parameter model of the MEGA software (version 11.0.13).

## Identification of *Ligia* spp. collected from ND and MD

Phylogenetic analysis of the mitochondrial 16S rRNA sequence showed that *Ligia* collected from ND and MD clustered distinctly with *L. exotica* and *L. cinerascens* retrieved from the GenBank (Fig 1). *Ligia oceanica* clustered as an outgroup. Owing to their morphological similarities, *L. exotica* and *L. cinerascens* were previously believed to be one species that existed in South Korea. Using mitochondrial 16S rRNA and 18S rRNA sequencing, Jung et al. [37] reported two species of *Ligia* in South Korea, which were divided into the "western" and "eastern" groups. In their study, the "western group" was dominant in the western coast, whereas the "eastern group" was abundant on the eastern coast and Jeju island. However, both were observed on the southern coast in different proportions depending on the location. In Masan, which is close to our sampling sites, ND and MD (located in Geoje-do), the levels of the "western group" were higher than those of "eastern group". Yin et al. [38] suggested that the "eastern group" sequences of South Korea were clustered with the *L. exotica* Roux (1828) clade, whereas the "western group" formed a clade with the *L. cinerascens* Budde-Lund (1885). Similar results were reported by Hurtado et al. [33] who conducted a phylogenetic analysis of the mitochondrial *16S rRNA*, *12S rRNA*, and *NaK* sequences of *Ligia* specimens from East Asia and 42 other locations worldwide. Along with these findings, specimens from ND were identified as *L. exotica*, and those from MD were identified as *L. cinerascens* in this study.

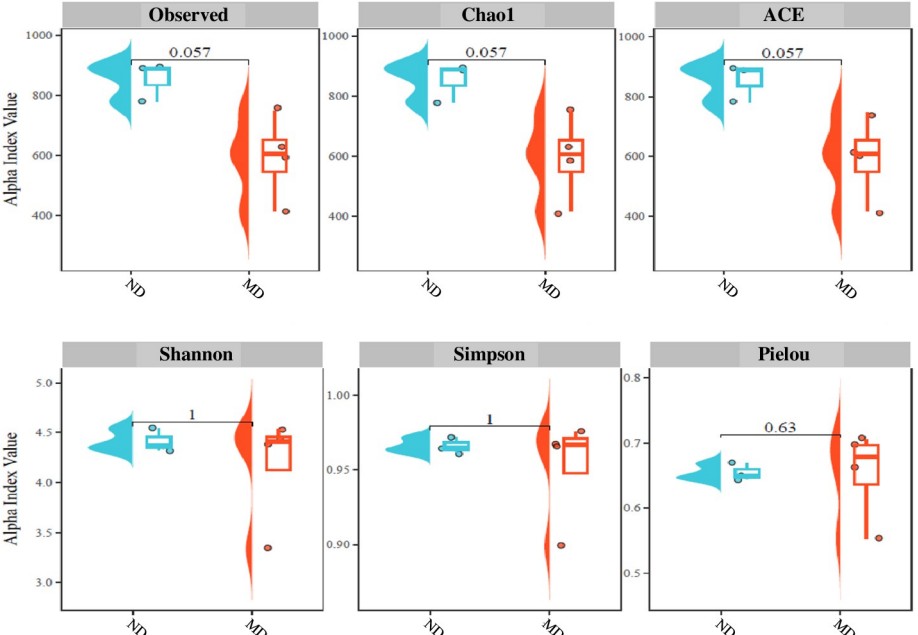

**Fig 2. Alpha diversity.** Observed species (operational taxonomic unit counts), Chao1, ACE, Shannon, Simpson, and pielou indices in individual samples from the ND and EPS-polluted MD sites.

## Gut bacterial communities of *Ligia* spp. collected from ND and MD

The total number of OTUs obtained from the 16S amplicon sequencing of *Ligia* gut was 779–894 in the ND group and 415–748 in the MD group. Alpha diversity in terms of Observed species, the Chao1, ACE, Shannon, Simpson, and Pielou indices, did not differ significantly between the two *Ligia* gut microbial communities collected from ND and MD, although richness (Observed species) was lower in MD than in ND (Fig 2). Similar results have been observed in humans [17], mice [16], marine medaka [39], and honey bees [40], where the number of Observed species decreased upon exposure to PET MPs exposure, suggesting that MPs can alter the structure of the gut microbiota.

Proteobacteria (67.47%, ND; 57.30%, MD), Bacteroidetes (13.63%, ND; 20.76%, MD), were the most abundant phyla detected at both sites, and followed by Firmicutes (5.00%, ND; 7.43%, MD), and Actinobacteriota (3.76%, ND; 5.63%, MD) (Fig 3). Desulfobacterota (4.81%) was specifically observed in *Ligia* from MD and Cyanobacteria (5.15%) in those from ND. In the terrestrial isopod, *Porcellionides pruinosus*, collected from three different sites in the Tunisia Harbor, Proteobacteria and Bacteroidetes were the representative phyla of all OTUs [41], which is similar to our observations. Firmicutes and Proteobacteria, which are involved in lignocellulose degradation in the gut of *P. pruinosus* in enrichment cultures, act as litter decomposers. In general, Bacteroidetes, Proteobacteria, Firmicutes, and Cyanobacteria are abundant in plastics [42], indicating their potential role in the degradation of MPs [43]. The 16S rRNA sequencing of bacterial communities in plastics collected from aquaculture regions also revealed that Proteobacteria, Cyanobacteria, Bacteroidetes, and Actinobacteria were dominant [44].

Unlike that at the phylum level, abundances at the family and genus levels differed between *Ligia* spp. collected from ND and MD (Table 1). At the family level, Rhodobacteraceae

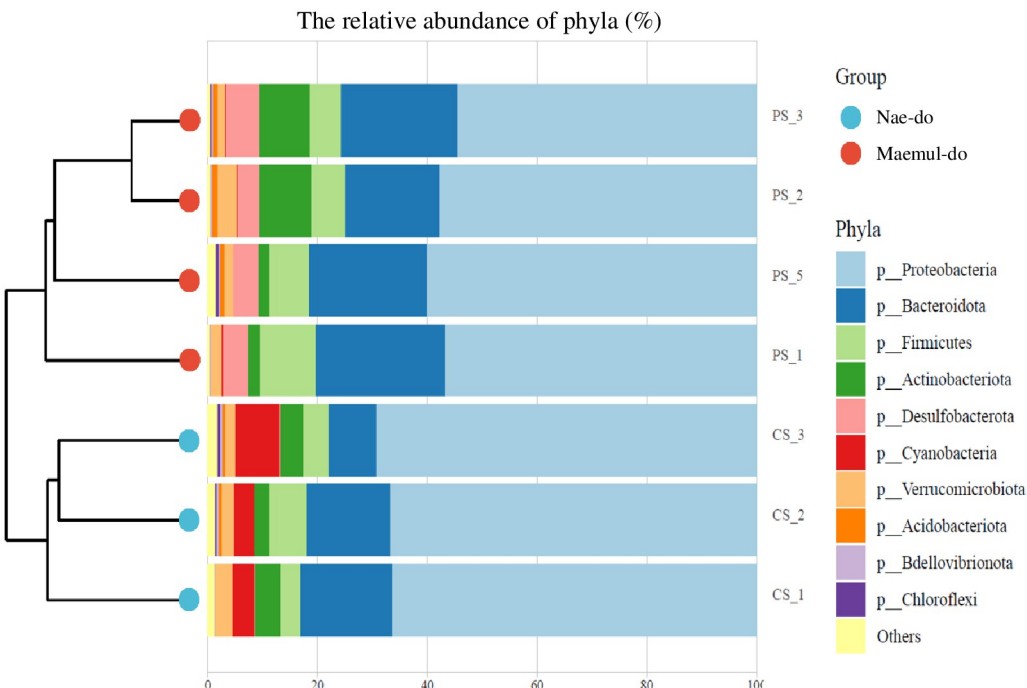

**Fig 3. Hierarchical clustering heatmap analyses.** The relative abundances of the top 10 bacterial phyla were detected using 16S amplicon sequencing from the ND and EPS-polluted MD sites.

**Table 1. Top five members from each taxonomic rank based on normalized proportion per sample.** Means (n = 3) ± standard error of mean are reported for each of the taxonomic assignments.

| | Clean site (ND) | Polluted site (MD) |
|---|---|---|
| **Family** | Rhodobacteraceae (27.0% ± 3.22) | Vibrionaceae (19.6% ± 9.61) |
| | uncult. Alphaproteobacteria (9.4% ± 4.45) | Rhodobacteraceae (17.2% ± 2.53) |
| | Vibrionaceae (9.3% ± 6.54) | Flavobacteriaceae (5.4% ± 1.86) |
| | uncult. Gammaproteobacteria (6.4% ± 3.39) | Desulfocapsaceae (4.3% ± 0.80) |
| | Flavobacteriaceae (5.8% ± 2.47) | Alphaproteobacteria (3.1% ± 0.78) |
| **Genus** | uncult. Rhodobacteraceae (20.1% ± 3.02) | *Vibrio* (19.6% ± 9.60) |
| | uncult. Alphaproteobacteria (9.4% ± 4.45) | uncult. Rhodobacteraceae (10.6% ± 2.67) |
| | *Vibrio* (9.3% ± 6.54) | *Candidatus* (4.9% ± 2.16) |
| | uncult. Gammaproteobacteria (6.4% ± 3.39) | *Roseimarinus* (4.7% ± 3.39) |
| | *Paracoccus* (4.5% ± 3.48) | *Desulfotalea* (4.3% ± 0.83) |

(26.98%), Alphaproteobacteria (9.38%), and Vibrionaceae (9.30%), belonging to the phylum Pseudomonadota, were abundant in *L. exotica* from ND, whereas Vibrionaceae (19.62%) and Rhodobacteraceae (17.20%) were more abundant in *L. cinerascens* from MD (Fig 4A and Table 1). Among the top 30 genera obtained from 16S amplicon sequencing, strains belonging to Rhodobacteraceae, *Vibrio* (Bacteriodotetes), and *Paracoccus* (Alphaproteobacteria) were abundant in *Ligia* from the ND, whereas *Vibrio* was the dominant taxon in *Ligia* from the MD, followed by strains belonging to Rhodobacteraceae, *Candidatus* (Bipolaricaulota), *Roseimarinus* (Bacteroidetes, Prolixibacteraceae), and *Desulfotalea* (Proteobacteria) (Fig 4B). LDA scores for assessing differentially abundant bacterial taxa between *Ligia* from ND and those from MD showed that strains belonging to Prolixibacteraceae (Bacteroidota), *Roseobacter* (Pseudomonadota), *Oceanisphaera* (Pseudomonadota), *Jejudonia* (Bacteriodota), *Demequina* (Actinomycetota), *Cryomorpha* (Bacteriodota), and *Brumimicrobium* (Bacteriodota) were significantly dominant at MD (Fig 5A). Analysis of the 16S rRNA sequence using shotgun sequencing revealed that the genera, *Vibiro* (Pseudomonadota), *Desulfobulbaceae* (Thermodesulfobacteriota), *Roseimarinus* (Bacteroidetes), and *Sunxiuqinia* (Bacteroidetes), were significantly more abundant in *L. cinerascens* from MD than in those from ND (Fig 5B). This suggests that the combined use of 16S rRNA amplicon sequencing and metagenomic analysis may be helpful in obtaining more information on the gut microbiome at the genus level.

Gram-negative bacterial strains belonging to the phyla Pseudomonas and Bacteroidetes isolated from soil samples may degrade plastic PE and PP by forming biofilms on both polymers [45]. For example, Rhodobacteraceae, belonging to Pseudomonadota, was predominant in completely developed plastic biofilms when the amount of enzymes required for carbohydrate hydrolysis increased significantly [46]. In particular, members of the family Rhodobacteraceae were found to be more abundant in plastic samples than in seawater, suggesting its role in degrading hydrocarbons, such as PE [44, 47]. Actinomycetes (predominantly *Streptomyces gougerotti*, *Micromonospora matsumotoense*, and *Nocardiopsis prasina*) derived from coastal sediments can degrade plastic films, such as low-density PE, PS, and polylactic acid [48].

The family Vibrionaceae is dominated by the genus *Vibrio*, which is commonly found in marine plastics [49]. Several reports have suggested that *Vibrio* spp. can degrade polymers of polyvinyl alcohol-linear low-density PE [50] and plastic bottle waste [51] as the sole carbon source. The proportion of symbiotic microbiota decreases and those of pathogens, and antibiotic-resistant and plastic-degrading microbes increase when MP levels increase in the gut [52]. Although many *Vibrio* spp. are harmless, they can cause serious damage to humans or animals [53]. The levels of *Vibrio* were higher in plastics in the ocean than in seawater samples, indicating that plastics can be potent vectors of pathogenic microorganisms [49, 54]. *Vibrio* spp. can degrade brown macroalgae with high carbohydrate content and are known to be alginate-metabolizing microorganisms [55]. Thus, several studies have recommended the use of *Vibrio* spp. as microbial platforms for biorefining brown macroalgae. These findings indicated that the high abundance of *Vibrio* in *Ligia* from MD may also be related to the digestion of seaweed and plastics as food sources in the present study. An increase in the number of opportunistic bacteria, such as *Vibrio* and *Rosimarinus*, in the water column is associated with oyster mortality [56]. The higher abundance of these pathogens in *Ligia* may underscore the unhygienic condition at EPS-polluted sites, which is related to the fewer OTUs in this group.

In general, the gut microbiota in vertebrates is influenced by the host's diet, habitat, lifestyle, and genetic factors [57, 58]. However, similar information in invertebrates is scarce. Although invertebrates are exposed to many microbes within their habitats, only a few bacterial species have been identified in their digestive tracts [59]. Recently, 16S amplicon sequencing of the gut microbiota of cephalopods revealed that diet and habitat contributed to the abundance of major gut microbes [59]. In the present study, the differences in the compositions of the gut

**A)**

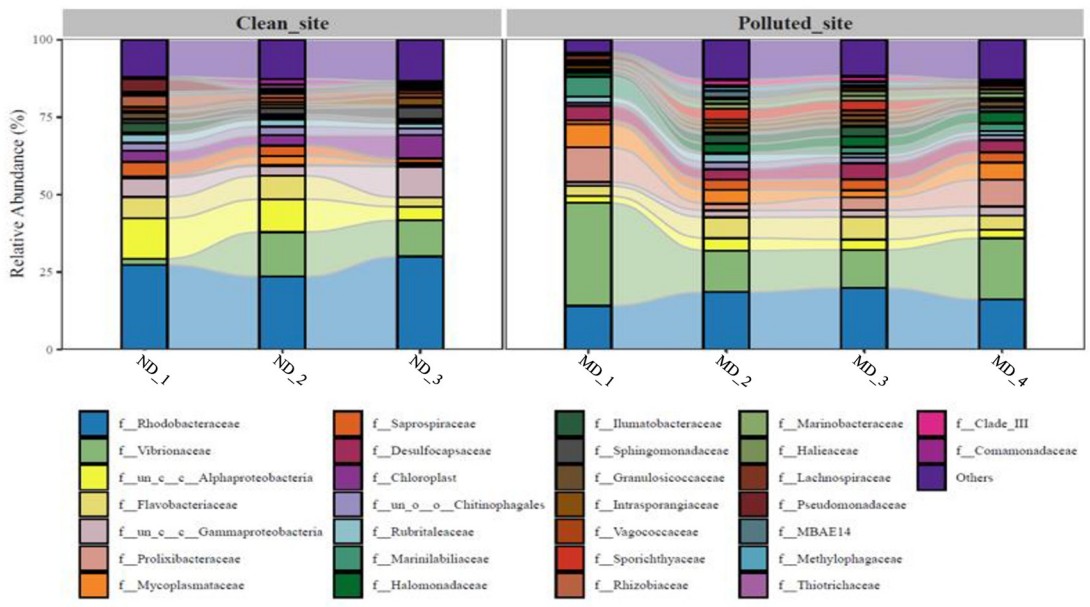

**B)**

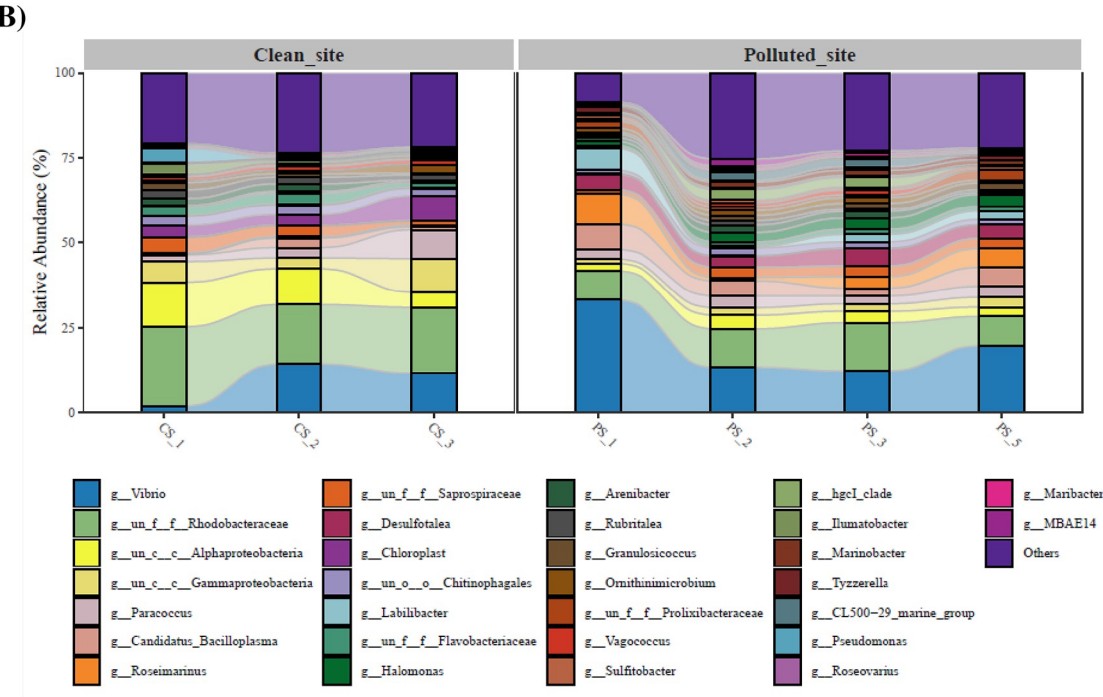

**Fig 4. Histogram of the relative abundance of the top 30 different bacterial taxonomic groups at the family level (A) and genus level (B) in the ND and EPS-polluted MD sites.** The y-axis shows the sequence percentage of each species in the total 16S rRNA sequences of each sample.

**(A)**

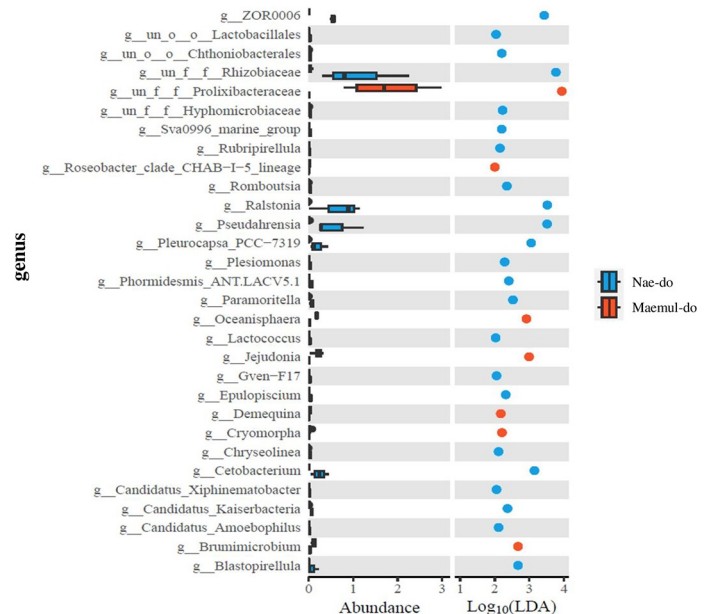

**(B)**

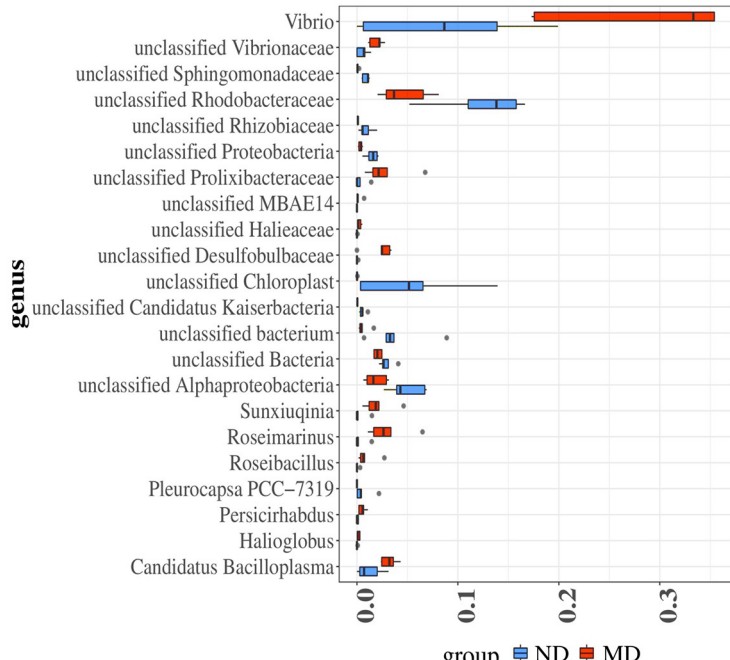

**Fig 5. Significantly different genera in ND and EPS-polluted MD sites.** A) Linear discriminant analysis (LDA) effect size (LEfSe) performed on the bacterial community relative abundance data (16s amplicon sequencing) and B) shotgun metagenome (percentage of 16S reads) in the ND and EPS-polluted MD sites. LDA scores were calculated using LDA effect size ($p < 0.05$ using Kruskal–Wallis test), using the linear discriminant analysis to assess the effect size of each differentially abundant bacterial taxa.

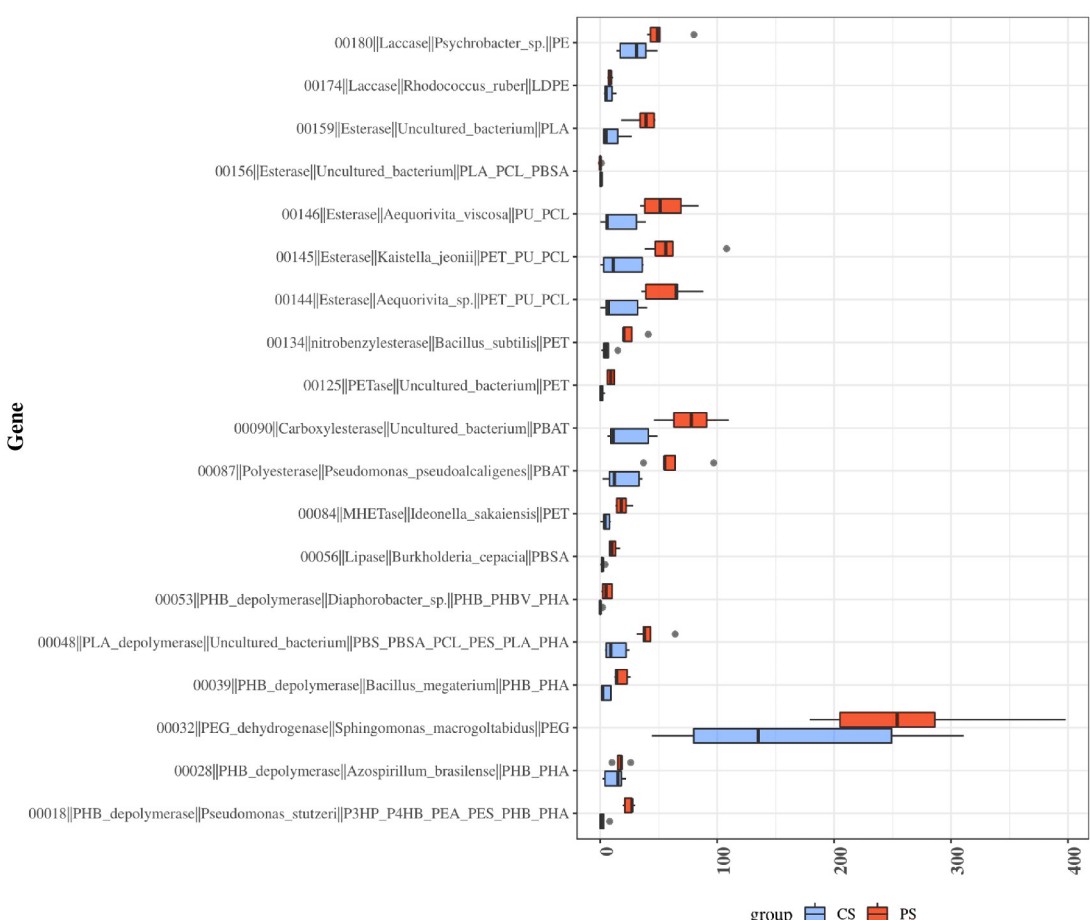

**Fig 6. Functional gene analysis using shotgun metagenome data and PlasticDB.** Boxplot showing significant differences in the abundance of plastic-degradation genes in microbes after comparing the groups of samples using DEseq2 ($p<0.05$).

microbiota in *Ligia* spp. may have been caused by varying habitat conditions, such as the higher levels of EPS at the MP site, which influenced the diet [29]. Thus, further information on the correlation between the gut microbiota and determining factors such as diet and habitat is required for these species.

In the environment, microorganisms can degrade plastics using plastic degradation enzymes (such as PETase, esterase, and cutinase) into organic compounds, which are utilized as carbon sources and finally released as carbon dioxide, accompanied by the production of higher-value bioproducts [60]. Microorganisms play a major role in the hydrocarbon cycle of oceans [42]. As the extent of plastic pollution increases in oceans, marine microorganisms may utilize plastic as a new carbon source [42]. Several studies have suggested that the gut microbiomes of some invertebrates, including insects and isopods, may be involved in the degradation of MPs [61, 62]. Functional gene analysis using shotgun metagenome data and PlasticDB showed that 19 plastic degradation-related genes and the bacteria-harboring them were present in the gut of *Ligia* (Fig 6 and Table 2) and were significantly more abundant in the EPS-polluted site (MD) than in the ND. The gene encoding polyethylene glycol (PEG)-dehydrogenase, which degrades PEG (monomer of plastic) and is produced by *Sphingomonas macrogoltabidus*, was the most abundant [63]. Next in abundance were genes encoding esterases (*Aequorivita viscosa*, *Kaistella jeonii*, *Aequorivita* spp.), carboxylesterase (uncultured

**Table 2. Top 10 plastic-degrading enzymes and microbes detected in *Ligia* from EPS-polluted MD.**

| Plastic-degrading enzyme | Protein ID | GenBank ID | Microorganism | Plastics |
|---|---|---|---|---|
| PHB depolymerase | 00018 | ACG63775.1 | *Stutzerimonas stutzeri* | P3HP, P4HB, PEA, PES, PHB, PHA |
| PEG dehydrogenase | 00032 | BAF98451.1 | *Sphingomonas macrogoltabidus* | polyethylene glycol |
| PLA depolymerase | 00048 | BAF57214.1 | Uncultured bacterium | PBS, PBSA, PCL, PES, PLA, PHA |
| Polyesterase | 00087 | AMW89397.1 | *Pseudomonas pseudoalcaligenes* | PBAT |
| Carboxylesterase | 00090 | AOR05750.1 | Uncultured bacterium | PBAT |
| P-nitrobenzylesterase | 00134 | ADH43200.1 | *Bacillus subtilis* | PET |
| Esterase | 00144 | WP_111881932 | *Aequorivita sp* | PET |
| | 00145 | WP_039353427 | *Kaistella jeonii* | PET, polyester polyurethane Impranil RDLN (PU), polycaprolactonate (PCL) |
| | 00146 | WP_073216622 | *Aequorivita viscosa* | polyester polyurethane Impranil RDLN (PU); polycaprolactonate (PCL) |
| Laccase | 00180 | UVG67878.1 | *Psychrobacter sp.* | Polyethylene |

bacteria), polyesterase (*P. pseudoalcaligenes*), and PLA-depolymerase (uncultured bacteria), known to be produced by strains belonging to Bacteroidetes and Proteobacteria. These enzymes contain esterase, hydrolase, and lipase domains involved in the degradation of lipophilic substances, indicating their role in plastic degradation. Indeed, they can be activated by various polymers, including PET, poly(butylene adipate-co-butylene terephthalate), poly lactic acid (PLA), and PEGs [64–66]. Prolixibacteracea (*Sunxiuquinia*), Flavobacteriaceae (*Aequorivita*), and Vibrionaceae (*Vibrio*) are the lignocellulolytic bacteria found in salt marshes that secrete lignocellulose-degrading enzymes (carbohydrate-active enzymes) [67]. Among these, esterases are involved in the decoupling of lignin from the polysaccharides in lignocellulose. These bacteria may also be involved in plastic decomposition because of the molecular and physico-chemical similarities between hydrophobic polymers, plastics, and lignocellulose, as suggested by Daly et al. [68]. In addition, the levels of PETase, lipase (*Burkholderia cepacia*), mono-(2-hydroxyethyl)terephthalic acid hydrolase (*Ideonella sakaiensis*), nitrobenzyl esterase (*Bacillus subtilis*), laccase (Phychrobacter, *Rhodococcus ruber*), and poly(3-hydroxybutyrate) depolymerase (*Bacillus megaterium*, *Azospirillum brasilense*, *Pseudomonas stutzeri*, *Diaphorobacter sp.*) were significantly higher in *Ligia* from the EPS-polluted site (MD) than in those from ND. These findings support the hypothesis that the gut bacterial communities are influenced by diet, including the presence of plastics. Several reports have suggested that environmental microorganisms, including species within the genera *Pseudomonas* and *Bacillus* spp., can degrade plastics using plastic degradation enzymes with different efficiencies [63, 69]. These findings indicated that the gut microbiota of *Ligia* from MD may be more potent in plastic ingestion and degradation than that of *Ligia* from ND. Interestingly, the most abundant enzyme, PEG-dehydrogenase, was also highly expressed in ND. Marine algal toxins possess a PEG-like structure [70]. PEG dehydrogenase has been reported to be involved in the degradation of PEG in *Flavobacterium* sp., which is associated with dinoflagellates [71]. Bacteria within *Roseobacter* isolated from coastal and oceanic environments can also transform algal toxins and lignin [72]. This indicates that the higher number of PEG dehydrogenase-coding genes observed in the guts of *Ligia* from both sites may have originated from algae. Indeed, algal blooms have been reported in the southern coastal waters of Korea [73].

KEGG enrichment analysis of the metagenome showed that the expression of most genes was significantly higher in *Ligia* from the EPS-polluted site (MD) than in those from the ND

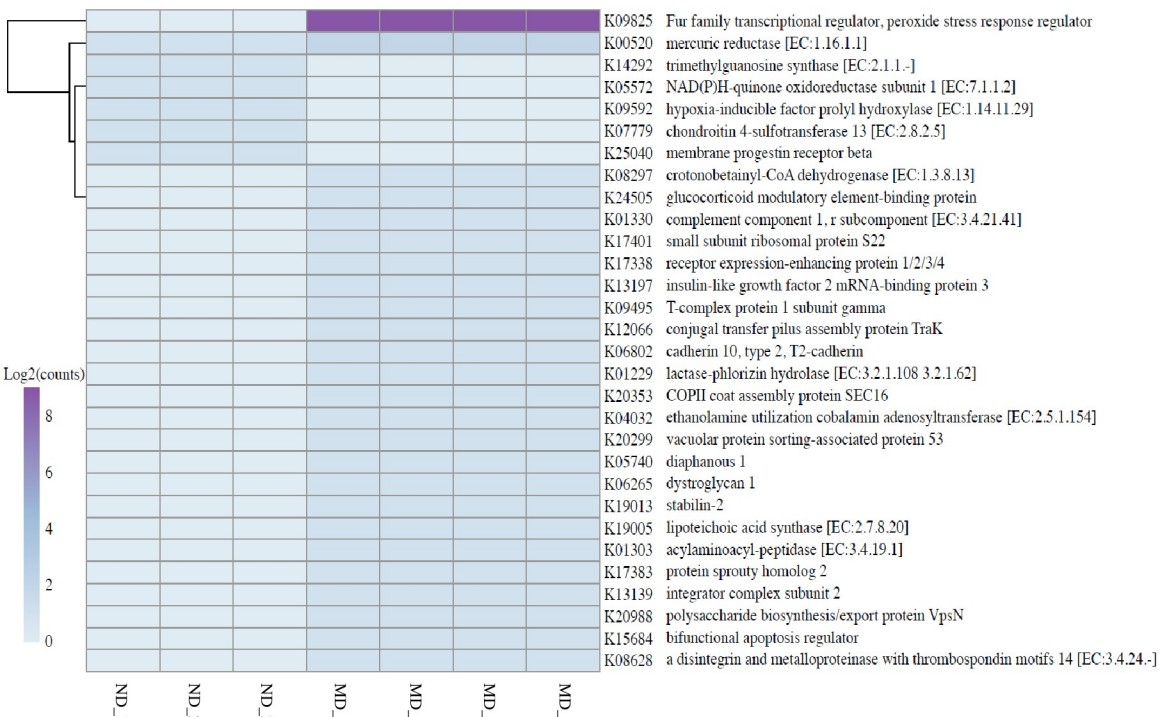

**Fig 7. Kyoto Encyclopedia of Genes and Genome (KEGG) pathways enrichment analysis of the metagenome.** Top 30 Significantly enriched KEGG pathways in the ND metagenomes relative to that in the MD metagenomes.

site, with the exception of gene encoding the numbers of five pathways (trimethylguanosine synthase, oxidoreductase subunit1, hypoxia-inducible factor, sulfotransferase, and progestin receptor) (Fig 7). The peroxide stress response regulator-related pathway (K09825), represented by peroxide resistance regulation repressor (perR), was most prominent in *Ligia* from the EPS-polluted site (MD). It plays an important role in the cellular defense system as a reactive oxygen species scavenger and protects bacteria against oxidative stress [74]. Leadbeater et al. [67] reported that oxidative enzymes, such as laccases, peroxidases, and lytic polysaccharide mono-oxygenases, are important for lignocellulose degradation in the terrestrial environment. During plastic degradation, the redox potential for electron extraction is higher than that for lignin degradation because of the hydrophobicity of plastics [75]. Most of the metabolism-related genes, such as those encoding crotonobetainyl-CoA dehydrogenase, lactase-phlorizin hydrolase, cobalamin adenosyltransferase, lipoteichoic acid synthase, acylaminoacyl-peptidase, and metalloproteinase, were more abundant in the EPS-polluted area (MD) than in the ND. Crotonobetainyl-CoA dehydrogenase belongs to the acyl-CoA dehydrogenase family and is involved in the downstream steps of PE degradation [76]. This indicate that the involvement of this pathway may be related to the high abundance of plastic debris at polluted sites (MD).

Lactase-phlorizin hydrolase is involved in the metabolism of polyphenols in the small intestine of humans [77], indicating the uptake of macroalgae as a food source by *Ligia*. Polyphenols protect against the oxidative stress induced by plastics and plastic additives [78]. This indicates that an increase in the activities of the polyphenol metabolic enzyme-related pathways may be associated with the protective role of the *Ligia* gut microbiome collected from EPS-polluted sites (MD). Acylaminoacyl peptidases protect against oxidative stress by removing damaged proteins [79].

Cobalamin adenosyltransferase participates in the biosynthesis of vitamin B12 (cobalamin), an essential vitamin, by bacterial communities via aerobic or anaerobic pathways [80]. Vitamin B12 is required by dehydratase, which is involved in the metabolism of ethylene glycol, a monomer of PET, polyethylene furanoate, and polyurethane (PU) [81]. Efforts to increase microbial vitamin B12 production have been made by the medical and food industries [82]. This implies a potential role for the *Ligia* gut microbiome in cobalamin biosynthesis. Metallo-proteinases are bacterial metal-containing proteases that contain metal ions, such as $Zn^{2+}$, $Mg^{2+}$, or $Cu^{2+}$ at their active sites [83]. These proteases are divided into endoproteases, which function in cells, and exoproteases, which are secreted outside the cells. Endoproteases activate water molecules during catalytic reactions, while exoproteases degrade environmental proteins to obtain nutrients from heterotrophic bacteria. Exoproteases are important virulence factors in pathogens [83].

Lipoteichoic acid (LTA) is an important cell wall polymer in Gram-positive bacteria and is found in the phylum Firmicutes, *Bacillus subtilis*, and *Staphylococcus aureus* [84]. LTA synthase plays a key role in cell wall biosynthesis [85]. This finding may be related to the higher abundance of Firmicutes in *Ligia* collected from MD than in those collected from ND (Table 1).

## Conclusion

In summary, we compared the gut microbial communities of *Ligia* collected from different sites using a metagenomic approach. The total OTUs of the *Ligia* gut microbiome were higher in ND than in EPS-contaminated MD; however, the relative abundances of phyla, families, and genera differed between locations. These findings suggest that the composition of the gut microbiome may be related to different habitats, environments, and diets, including the occurrence of plastic pollution. Functional gene analysis detected 19 plastic degradation-related genes, such as those encoding PEG-hydrogenase, esterase, and carboxylesterase, in the gut microbes of *Ligia* from EPS-contaminated MD, indicating the potential role of the *Ligia* gut microbiome in plastic degradation. KEGG analysis of the *Ligia* gut bacterial genome showed that the significant increase in metabolism-related pathways may be associated with the degradation of carbon sources, including plastics, oxidative stress defenses, and nutrient biosynthesis. Despite the limitations of metagenomic analysis at the species level, these metagenomic data will be useful as a scientific basis for elucidating the functions of *Ligia* gut bacteria in further studies. However, further studies on the correlation between the gut microbiota and environmental factors, such as diet and habitat, and the plastic-degrading capability of the gut microbiota are required for these species. Overall, this study improves our understanding regarding the dynamics of the gut microbiome community in *Ligia* under different environmental conditions of MP pollution.

## Supporting information

**S1 Fig. (a) Two sampling locations of *Ligia* sp., and the number of microplastic (MP) contamination per individual organism (b) Nae-do (34°38'47.30"N 128°34'39.00"E), Geoje and (c) Maemul-do (34°47'21.50"N 128°42'58.10"E), Geoje.** The map image was created using a free, open-source base map program (the QGIS: QGIS is an open-source GIS tool) (Public Data Portal URL: https://www.data.go.kr/data/3035495/fileData.do) (Choi et al. [26]). (DOCX)

## Author Contributions

**Data curation:** Young-Mi Lee.

**Formal analysis:** Kwang-Min Choi.

**Funding acquisition:** Jee-Hyun Jung.

**Investigation:** Kwang-Min Choi, Seong Hee Mun, Je-Won Yoo.

**Methodology:** Je-Won Yoo.

**Software:** Je-Won Yoo.

**Supervision:** Jee-Hyun Jung.

**Validation:** Je-Won Yoo.

**Visualization:** Young-Mi Lee, Kwang-Min Choi, Seong Hee Mun.

**Writing – original draft:** Jee-Hyun Jung.

**Writing – review & editing:** Young-Mi Lee, Jee-Hyun Jung.

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
