## [Decision Letter · Decision Letter 0]

7 Jun 2024

PONE-D-24-19582Gut microbiota composition of the Isopod Ligia in South Korea exposed to expanded polystyrene pollution  PLOS ONE

Dear Dr. Jung,

Thank you for submitting your manuscript to PLOS ONE. After careful consideration, we feel that it has merit but does not fully meet PLOS ONE’s publication criteria as it currently stands. Therefore, we invite you to submit a revised version of the manuscript that addresses the points raised during the review process.

We look forward to receiving your revised manuscript.

Kind regards,

Sinosh Skarlyachan, PhD

Academic Editor

PLOS ONE

Journal Requirements:

"This work was supported by the grants “Risk assessment to prepare standards for protecting marine ecosystem(KIMST-20230383)”and “Land and sea based input and fate of microplastics in the marine environment (KIMST-20230383)of the Korea Institute of Marine Science & Technology Promotion (KIMST) funded by the Ministry of Oceans and Fisheries”, and the National Research Foundation of Korea(NRF) grant funded by the Korea government(MSIT) (No. 2023R1A2C1005630)."

5. Please note that funding information should not appear in the Acknowledgments section or other areas of your manuscript. We will only publish funding information present in the Funding Statement section of the online submission form. Please remove any funding-related text from the manuscript. 

6. Please provide a complete Data Availability Statement in the submission form, ensuring you include all necessary access information or a reason for why you are unable to make your data freely accessible. If your research concerns only data provided within your submission, please write "All data are in the manuscript and/or supporting information files" as your Data Availability Statement.

7. PLOS requires an ORCID iD for the corresponding author in Editorial Manager on papers submitted after December 6th, 2016. Please ensure that you have an ORCID iD and that it is validated in Editorial Manager. To do this, go to ‘Update my Information’ (in the upper left-hand corner of the main menu), and click on the Fetch/Validate link next to the ORCID field. This will take you to the ORCID site and allow you to create a new iD or authenticate a pre-existing iD in Editorial Manager. Please see the following video for instructions on linking an ORCID iD to your Editorial Manager account: https://www.youtube.com/watch?v=_xcclfuvtxQ

8. Please include your full ethics statement in the ‘Methods’ section of your manuscript file. In your statement, please include the full name of the IRB or ethics committee who approved or waived your study, as well as whether or not you obtained informed written or verbal consent. If consent was waived for your study, please include this information in your statement as well. 

9. We note that Supporting Figure S1 in your submission contain map images which may be copyrighted. All PLOS content is published under the Creative Commons Attribution License (CC BY 4.0), which means that the manuscript, images, and Supporting Information files will be freely available online, and any third party is permitted to access, download, copy, distribute, and use these materials in any way, even commercially, with proper attribution. For these reasons, we cannot publish previously copyrighted maps or satellite images created using proprietary data, such as Google software (Google Maps, Street View, and Earth). For more information, see our copyright guidelines: http://journals.plos.org/plosone/s/licenses-and-copyright.

1) You may seek permission from the original copyright holder of Supporting Figure S1 to publish the content specifically under the CC BY 4.0 license.  

2) If you are unable to obtain permission from the original copyright holder to publish these figures under the CC BY 4.0 license or if the copyright holder’s requirements are incompatible with the CC BY 4.0 license, please either i) remove the figure or ii) supply a replacement figure that complies with the CC BY 4.0 license. Please check copyright information on all replacement figures and update the figure caption with source information. If applicable, please specify in the figure caption text when a figure is similar but not identical to the original image and is therefore for illustrative purposes only.

Reviewers' comments:

Reviewer's Responses to Questions

**Comments to the Author**

1. Is the manuscript technically sound, and do the data support the conclusions?

Reviewer #1: Yes

Reviewer #2: Partly

Reviewer #3: Partly

2. Has the statistical analysis been performed appropriately and rigorously? 

Reviewer #1: Yes

Reviewer #2: Yes

Reviewer #3: Yes

3. Have the authors made all data underlying the findings in their manuscript fully available?

Reviewer #1: Yes

Reviewer #2: Yes

Reviewer #3: Yes

4. Is the manuscript presented in an intelligible fashion and written in standard English?

Reviewer #1: Yes

Reviewer #2: Yes

Reviewer #3: No

5. Review Comments to the Author

Reviewer #1: The manuscript is well written. The methodology used for identification, extraction and metagenomic sequencing of gut microbiota is well described with all the necessary details.

I would recommend adding the limitations of the metagenomic approach used the study in the conclusion section to make the manuscript well rounded.

Reviewer #2: Ln 73-77. Are there more evidences suggesting the involvement of isopod gut microbiome in plastic degradation?

Ln99. How many samples were collected in total from the collection site? How many of these samples were utilized for further analysis and what was the criteria for choosing those samples?

Ln 111. Has the same amount of template DNA been added into each of the reactions?

Ln121. How were the default parameters determined?

Ln 140. Have the authors used other RNA seq analysis tools to obtain more comprehensive data and avoid tool specific bias?

Ln 189. How was the quantity of plastic debris measured? How much area was used for measurement? Is the measurement consistent throughout the region and how is the isopod population affected by amount of debris found? Is it a linear correlation?

Reviewer #3: The present manuscript assesses the role of Ligia sp. in plastic degradation and examines the potential impacts of microplastics on microbiome composition. This issue is of significant importance and interest, with considerable potential implications. However, several concerns must be addressed before publication:

Major Comments:

-The authors describe "clean site" and "polluted site" for the collection of specimens in the materials and methods section. How did the authors characterize these sampling locations? This should be included in the materials and methods section. Considering that the authors have previously characterized the plastic debris in these areas in earlier studies, they should reference these studies in the materials and methods section.

-The authors make important statements, such as in line 383: "Several reports have suggested that environmental microorganisms, including species within the genera Pseudomonas and Bacillus spp., can degrade plastics using plastic degradation enzymes with different efficiencies [63, 69]. These findings indicated that the gut microbiota of Ligia from MD may be more potent in plastic ingestion and degradation than that of Ligia from ND." The authors assume that the presence of certain taxonomic groups (at least at the genus level) indicates a higher potential to degrade plastics. Such statements, which are found throughout the manuscript, are highly speculative. The authors cannot assume that all members of the genus Vibrio or Bacillus participate in plastic degradation. It can be more acceptable if the authors increase the power of the 16s analysis and reach to species level, but they should point out that those species are known for having potential to degrade plastics, and if the enrichment of this species is due to higher amount of plastics in that environment would require further study.

-Although some pathways from the microbiota are highlighted in figure 6, which is a key figure for this study, I strongly suggest that the authors perform a PICRUSt analysis to identify which metabolic pathways are enriched and associated with the microbiota. The results should be discussed alongside those in figure 6 to better understand other potential relationships withing the microbiota metabolism.

-The biggest concern of this reviewer is the considerable variability that can be produced by multiple factors in this study, which should not be ignored. Throughout the manuscript, the authors emphasize the critical changes in the microbiome due to microplastics. However, different sampling areas, species, and conditions are used. In the conclusions, the authors state, "These findings suggest that the composition of the gut microbiome may be related to different habitats, environments, and diets, including the occurrence of plastic pollution." I strongly suggest that the authors revise the manuscript to focus on this conclusion instead. Considering this, the study loses strength. I suggest that the authors conduct experiments under controlled conditions, acclimatizing the Ligia and using a commercially available mix of microplastics and expanded polystyrene. Such results would clarify whether the observed changes are due to microplastics or other environmental conditions.

-Similarly, I believe the study would benefit from additional in vitro experiments isolating the most interesting microbiota with potential to degrade microplastics and demonstrating this potential in vitro. If possible, metatranscriptome analysis should be conducted to identify which pathways change in the presence of these plastics.

Minor Comments:

-Lines 72-74: I suggest introducing arthropods, such as insects, instead of directly discussing insects. This is confusing considering that the study model is a crustacean.

-Have the authors assessed whether microplastics have any potential deleterious effects on any Ligia species? Or whether the presence of any Ligia species in non-contaminated areas and their absence in contaminated areas can be associated with microplastics?

6. PLOS authors have the option to publish the peer review history of their article (what does this mean?). If published, this will include your full peer review and any attached files.

Reviewer #1: No

Reviewer #2: No

Reviewer #3: **Yes: **Alvaro Fernandez-Montero

---

## [Author Response · Author response to Decision Letter 0]

2 Jul 2024

Responses to Reviewers’ Comments

Manuscript No.: PONE-D-24-19582

RESPONSES TO COMMENTS

We greatly appreciate the critical comments of the reviewers on our work. According to the reviewer’s comments, we thoroughly revised our manuscript and incorporated changes as suggested by the reviewer (please, check the red color in the revised manuscript). We answered one by one to comments raised by reviewers as follows:

Journal Requirements:

Response: We confirmed that this revised manuscript meets PLOS ONE’s style requirements, such as file naming for supple figures, Titles, and affiliations according to the author’s guidelines.

Response: We added the information regarding the permits of sampling locations in the method. “this area was an area where there was no permitting procedure required for marine Ligia spp. collection; thus, it was legal to freely sample. (Line 97-98)

Response: We removed “Funding information” from the revised manuscript and added a “financial disclosure” file. 

"This work was supported by the grants “Risk assessment to prepare standards for protecting marine ecosystem(KIMST-20230383)”and “Land and sea based input and fate of microplastics in the marine environment (KIMST-20230383)of the Korea Institute of Marine Science & Technology Promotion (KIMST) funded by the Ministry of Oceans and Fisheries”, and the National Research Foundation of Korea(NRF) grant funded by the Korea government(MSIT) (No. 2023R1A2C1005630)."

Response: We added the funder’s role in the financial disclosure file. 

5. Please note that funding information should not appear in the Acknowledgments section or other areas of your manuscript. We will only publish funding information present in the Funding Statement section of the online submission form. Please remove any funding-related text from the manuscript. 

Response: We removed the funding information in the revised manuscript.

6. Please provide a complete Data Availability Statement in the submission form, ensuring you include all necessary access information or a reason for why you are unable to make your data freely accessible. If your research concerns only data provided within your submission, please write "All data are in the manuscript and/or supporting information files" as your Data Availability Statement.

Response: We provide a complete Data Availability Statement in the revised manuscript. 

"All data are in the manuscript and/or supporting information files" as your Data Availability Statement.” (Line 467)

7. PLOS requires an ORCID iD for the corresponding author in Editorial Manager on papers submitted after December 6th, 2016. Please ensure that you have an ORCID iD and that it is validated in Editorial Manager. To do this, go to ‘Update my Information’ (in the upper left-hand corner of the main menu), and click on the Fetch/Validate link next to the ORCID field. This will take you to the ORCID site and allow you to create a new iD or authenticate a pre-existing iD in Editorial Manager. Please see the following video for instructions on linking an ORCID iD to your Editorial Manager account: https://www.youtube.com/watch?v=_xcclfuvtxQ

Response: The corresponding author has been ORCID ID and checked its validation.

8. Please include your full ethics statement in the ‘Methods’ section of your manuscript file. In your statement, please include the full name of the IRB or ethics committee who approved or waived your study, as well as whether or not you obtained informed written or verbal consent. If consent was waived for your study, please include this information in your statement as well. 

Response: Since arthropods, including Ligia spp., do not require permission from our institution's IRB committee, they have not been approved in this study. Tissues were separated by liquid nitrogen freezing immediately after field collection.

9. We note that Supporting Figure S1 in your submission contain map images which may be copyrighted. All PLOS content is published under the Creative Commons Attribution License (CC BY 4.0), which means that the manuscript, images, and Supporting Information files will be freely available online, and any third party is permitted to access, download, copy, distribute, and use these materials in any way, even commercially, with proper attribution. For these reasons, we cannot publish previously copyrighted maps or satellite images created using proprietary data, such as Google software (Google Maps, Street View, and Earth). For more information, see our copyright guidelines: http://journals.plos.org/plosone/s/licenses-and-copyright.

 1) You may seek permission from the original copyright holder of Supporting Figure S1 to publish the content specifically under the CC BY 4.0 license. 

2) If you are unable to obtain permission from the original copyright holder to publish these figures under the CC BY 4.0 license or if the copyright holder’s requirements are incompatible with the CC BY 4.0 license, please either i) remove the figure or ii) supply a replacement figure that complies with the CC BY 4.0 license. Please check copyright information on all replacement figures and update the figure caption with source information. If applicable, please specify in the figure caption text when a figure is similar but not identical to the original image and is therefore for illustrative purposes only.

Response: Our Supporting Figure S1 image was created using a free, open-source base map program (the QGIS: QGIS is an open-source GIS tool) (Public Data Portal URL: https://www.data.go.kr/data/3035495/fileData.do), and there is no copyright problem. We attached the copyright information of the map image in the revision and added website information on the legend of Figure S1 in the revised manuscript (Line 720-722) 

[Information of the map image]

 

Reviewers' comments:

Reviewer's Responses to Questions

Comments to the Author

1. Is the manuscript technically sound, and do the data support the conclusions?

Reviewer #1: Yes

Reviewer #2: Partly

Reviewer #3: Partly

Response: We also checked the conclusion and corrected it based on the data. 

2. Has the statistical analysis been performed appropriately and rigorously?

Reviewer #1: Yes

Reviewer #2: Yes

Reviewer #3: Yes

Response: We had appropriately performed the statistical analysis. 

3. Have the authors made all data underlying the findings in their manuscript fully available?

Reviewer #1: Yes

Reviewer #2: Yes

Reviewer #3: Yes

Response: Our manuscript included all of the data from the study.

4. Is the manuscript presented in an intelligible fashion and written in standard English?

Reviewer #1: Yes

Reviewer #2: Yes

Reviewer #3: No

Response: We also checked any typographical or grammatical errors in our manuscript, which were re-corrected from Cactus Communications Inc. (214 Carnegie Center, Suite 102, Princeton, NJ 08540, USA).

5. Review Comments to the Author

Reviewer #1: The manuscript is well written. The methodology used for identification, extraction and metagenomic sequencing of gut microbiota is well described with all the necessary details.

I would recommend adding the limitations of the metagenomic approach used the study in the conclusion section to make the manuscript well rounded.

Response: We added the limitations of the metagenomic approach and the requirement for further study in conclusion. (Line 458-462)

Reviewer #2: 

Ln 73-77. Are there more evidences suggesting the involvement of isopod gut microbiome in plastic degradation?

Response: We additionally analyzed the publication database relating the isopod gut microbiome in plastic degradation but we could not find out the research report.

Ln99. How many samples were collected in total from the collection site? How many of these samples were utilized for further analysis and what was the criteria for choosing those samples?

Response: This study is a part of large-scale research “Effects of EPS on Ligia using multi-omics approach and isolation of plastic degrading bacteria from the gut of Ligia”. The parallel studies included plastic/chemical analysis, transcriptome analysis, and microbiome analysis. For this study, we used thirty wharf roaches from the total 100 individuals collected, but the sample size was reduced to ten individuals due to difficulties in the DNA extraction and screening process, such as body length (23.02 ± 1.17 mm) and body weight (0.70 ± 0.12 g) in order to minimize the difference between individuals (Lines 93-94). However, we tried to analyze more than 3 replicates at each site for statistical analysis. Since this study is a field study, sampling is always a challenge. 

Ln 111. Has the same amount of template DNA been added into each of the reactions?

Response: Although it was not necessary to match the amount of DNA in equal quantities for amplification of the 16S rRNA gene, we used the 10 ng of DNA sample concentration for PCR analysis. We revised it (Line 108)

Ln121. How were the default parameters determined?

Response: The default parameters in GBLOCKs v.0.91b are as follows:

After alignment, ambiguous regions (i.e. containing gaps and/or poorly aligned) were removed with Gblocks (v0.91b) using the following parameters: minimum length of a block after gap cleaning: 5; positions with a gap in less than 50% of the sequences were selected in the final alignment if they were within an appropriate block; all segments with contiguous nonconserved positions bigger than 4 were rejected; and minimum number of sequences for a flank position: 55%

Ln 140. Have the authors used other RNA seq analysis tools to obtain more comprehensive data and avoid tool specific bias?

Response: Our study only covered results of mitochondrial 16S rRNA and metagenome shotgun sequencing analysis in the wharf roach gut. Therefore, we did not show any RNA-seq analysis data. In our previous manuscript, we showed the RNA-seq results in wharf roaches (Choi et al., 2023). 

Choi Y, Shin D, Hong CP, Shin DM, Cho SH, Kim SS, et al. The effects of environmental Microplastic on wharf roach (Ligia exotica): A Multi-Omics approach. Chemosphere 2023;335:139122.

Ln 189. How was the quantity of plastic debris measured? How much area was used for measurement? Is the measurement consistent throughout the region and how is the isopod population affected by amount of debris found? Is it a linear correlation?

Response: In our previous results, we identified the MPs (by plastic size, type, and shape) using the micro-Fourier transform infrared microscope (μ Thermo Nicolet 6700 Continue μ FTIR; m; Thermo Scientific, Waltham, MA, USA). The 80% region of area was measured in our study. The number of particles was 1 particle in blank and reference site. The effect of population is not this research content, but the potential effects depend on microplastic contamination presented in our previous study (Choi et al., 2023). This study provides the first comparative field evidence of the gut microbiota dynamics of plastic detritus consumers in marine ecosystems.

Choi Y, Shin D, Hong CP, Shin DM, Cho SH, Kim SS, et al. The effects of environmental Microplastic on wharf roach (Ligia exotica): A Multi-Omics approach. Chemosphere 2023;335:139122.

Reviewer #3: The present manuscript assesses the role of Ligia sp. in plastic degradation and examines the potential impacts of microplastics on microbiome composition. This issue is of significant importance and interest, with considerable potential implications. However, several concerns must be addressed before publication:

Major Comments:

1) The authors describe "clean site" and "polluted site" for the collection of specimens in the materials and methods section. How did the authors characterize these sampling locations? This should be included in the materials and methods section. Considering that the authors have previously characterized the plastic debri

---

## [Decision Letter · Decision Letter 1]

22 Jul 2024

Gut microbiota composition of the Isopod Ligia in South Korea exposed to expanded polystyrene pollution

PONE-D-24-19582R1

Dear Dr. Jung,

We’re pleased to inform you that your manuscript has been judged scientifically suitable for publication and will be formally accepted for publication once it meets all outstanding technical requirements.

Kind regards,

Sinosh Skarlyachan, PhD

Academic Editor

PLOS ONE

Additional Editor Comments (optional):

Reviewers' comments:

Reviewer's Responses to Questions

**Comments to the Author**

1. If the authors have adequately addressed your comments raised in a previous round of review and you feel that this manuscript is now acceptable for publication, you may indicate that here to bypass the “Comments to the Author” section, enter your conflict of interest statement in the “Confidential to Editor” section, and submit your "Accept" recommendation.

Reviewer #1: All comments have been addressed

Reviewer #2: All comments have been addressed

2. Is the manuscript technically sound, and do the data support the conclusions?

Reviewer #1: Yes

Reviewer #2: Yes

3. Has the statistical analysis been performed appropriately and rigorously? 

Reviewer #1: Yes

Reviewer #2: Yes

4. Have the authors made all data underlying the findings in their manuscript fully available?

Reviewer #1: Yes

Reviewer #2: Yes

5. Is the manuscript presented in an intelligible fashion and written in standard English?

Reviewer #1: Yes

Reviewer #2: Yes

6. Review Comments to the Author

Reviewer #1: (No Response)

Reviewer #2: (No Response)

7. PLOS authors have the option to publish the peer review history of their article (what does this mean?). If published, this will include your full peer review and any attached files.

Reviewer #1: No

Reviewer #2: No

---

## [Editor Report · Acceptance letter]

26 Jul 2024

PONE-D-24-19582R1 

PLOS ONE

Dear Dr. Jung, 

I'm pleased to inform you that your manuscript has been deemed suitable for publication in PLOS ONE. Congratulations! Your manuscript is now being handed over to our production team.

Kind regards, 

on behalf of

Dr. Sinosh Skarlyachan 

Academic Editor

PLOS ONE